# Towards More Sustainable Meat Products: Extenders as a Way of Reducing Meat Content

**DOI:** 10.3390/foods9081044

**Published:** 2020-08-03

**Authors:** Tatiana Pintado, Gonzalo Delgado-Pando

**Affiliations:** 1Institute of Food Science, Technology and Nutrition (CSIC), José Antonio Novais 10, 28040 Madrid, Spain; tatianap@ictan.csic.es; 2Teagasc Ashtown Food Research Centre, Dublin, D15 DY05, Ireland

**Keywords:** meat extenders, meat products, meat substitutes, sustainability, plant-based proteins, insects, by-products, pulses, mushrooms

## Abstract

The low efficiency of animal protein (meat products) production is one of the main concerns for sustainable food production. However, meat provides high-quality protein among other compounds such as minerals or vitamins. The use of meat extenders, non-meat substances with high protein content, to partially replace meat, offers interesting opportunities towards the reformulation of healthier and more sustainable meat products. The objective of this review is to give a general point of view on what type of compounds are used as meat extenders and how they affect the physicochemical and sensory properties of reformulated products. Plant-based ingredients (pulses, cereals, tubers and fruits) have been widely used to replace up to 50% of meat. Mushrooms allow for higher proportions of meat substitution, with adequate results in reduced-sodium reformulated products. Insects and by-products from the food industry are novel approaches that present an opportunity to develop more sustainable meat products. In general, the use of meat extenders improves the yield of the products, with slight sensory modifications. These multiple possibilities make meat extenders’ use the most viable and interesting approach towards the production of healthier meat products with less environmental impact.

## 1. Introduction

In 2015, all United Nation Member States adopted “The 2030 Agenda for Sustainable Development” [1]. In this agenda, the countries agreed to 17 Sustainable Development Goals (SDG) to be achieved by the end of 2030. Sustainable food production is one of the main pillars of the document, where foods needs to be sufficient, safe, affordable and nutritious, as well as part of a sustainable production system. The world population growth and industrial development are causing an expansion of food production and an increased demand for animal protein [2]. One of the main concerns is the low efficiency of animal protein production. It is estimated that 7 kg of food from plant origin (animal feed) yields 1 kg of milk or meat for human consumption [3]. In addition, animal production is believed to use around 30% of the global land surface, contributing to deforestation and the loss of biodiversity [4]. However, the environmental impact of livestock production goes further than biodiversity loss: important greenhouse gas emissions, vast use of fertilisers and the deterioration of water quality due to effluents [4,5]. Westhoek et al. [6] estimated that “halving the consumption of meat, dairy products and eggs in the European Union would achieve a 40% reduction in nitrogen emissions, 25–40% reduction in greenhouse gas emissions and 23% per capita less use of cropland for food production”. However, meat represents an important source of energy, high-quality protein and micronutrients such as iron, zinc, selenium, vitamin B12 and vitamin D [5,7]. Meat and meat products currently provide one-sixth of the total energy intake of a European adult and widely contribute to total protein, vitamin D and iron consumptions up to 40%, 30% and 23%, respectively [7]. Hence, meat and meat products should not be disregarded in the diet, as they contribute to the avoidance of essential nutrient deficiencies and can also protect against malnutrition in countries where access to other types of highly nutritious products is limited [5,8]. Deficiencies of iron and vitamin D are of high prevalence around the world [9,10]. A suboptimal vitamin B12 status occurs in 30–60% of the population, mainly in less-developed countries [11]. A recent study by Vatanparast et al. [12] found that decreasing by 50% the red and processed meat consumption and increasing by 100% the consumption of plant-based alternatives in Canadian individuals improved the overall nutritional diet value but adversely affected the intake of protein, zinc and vitamin B12. However, not only undeveloped or developing countries are affected. Rippin et al. [8] detected deficiencies in these micronutrients for certain segments of the European population. There is no unique food alternative to meat or meat products with similar nutritional profiles, and even a combination of several foods does not assure the same nutritional intake. Vitamin B12 is only present in foods of animal origin, which makes people following vegan and vegetarian diets in need of supplementations to achieve the dietary reference intake (DRI) for this micronutrient [13]. Furthermore, non-meat foods contain only 20–60% the protein density of that of the meat, and the digestibility and bioavailability of some micronutrients from these sources are known to be lower [14]. Even though highly desirable, a vast improvement of meat production efficiency and sustainability in the near future is not likely. Current strategies should focus on limiting the environmental impact of our diets without risking nutrition deficiencies.

Meat products are inherent to food culture and are widely consumed all around the world. Imamura et al. [15] estimated a global consumption of processed meat from 3.9 g/day (first quintile) to 34 g/day (fifth quintile). Even though their consumption has been linked with the burden of chronic diseases like coronary heart disease, type 2 diabetes and certain types of cancer [16,17], a number of gaps still exist (such as the underlying mechanisms of cancer development and the role of cooking) that could offer room for mitigation during their processing [16,18]. Versatility, more attractive products, waste reduction opportunities and higher shelf lives are some core characteristics that differentiate meat products from fresh meat. Therefore, the reformulation of meat products to produce healthier and more sustainable versions seems like a robust strategy in-line with the SDG. The research on the development of healthier meat products started in the 1990s but still comprises a big proportion of the current research in this field. Two strategies are primarily followed: the reduction of harmful components to appropriate amounts and the incorporation of potentially health-enhancing ingredients [19]. The former is focused on the reduction of harmful saturated fatty acids [20], salt [21], cholesterol [22] and additives such as nitrite [23] or phosphates [24], whereas the latter studies the incorporation of the so-called “functional ingredients”, mainly from plant origin, that provide healthier characteristics to the product [25,26,27].

In the last decade, meat substitutes or analogues have received much interest as plant-based similar-in-properties alternatives to conventional meat products [28]. However, most of these analogues are produced under heavy processing manufacture and, thus, limiting the environmental sustainability gain and losing the healthier prerogative that they were originally based on. Meat reduction arises as a more meaningful alternative to a complete elimination of meat from the diet and, sometimes, a more sustainable option than meat substitutes [29,30]. The integration of plant-based ingredients into meat dishes has been proven as a successful and consumer-accepted strategy [29] and has opened the way to a different approach towards the reformulation of healthier and more sustainable meat products: meat substitutions with plant-based ingredients. Although originally devised to reduce costs, the use of meat extenders presents an opportunity to reduce the meat content while incorporating some healthier ingredients to the meat product. Meat extenders are non-meat substances with high protein contents that can also modify some of the product’s properties, such as water-holding capacity (WHC), texture, palatability and appearance [31].

In this review, we aim to evaluate the use of extenders as meat substitutes and how they affect the physicochemical and sensory properties of the meat products. The review has been structured in different sections, where meat extenders have been grouped based on their origin. In addition, a section discussing the consumer perspective about the acceptance of novel and more sustainable meat products has also been included.

## 2. Meat Extenders

### 2.1. Pulses as Meat Extenders

“Pulses are edible dry seeds of plants belonging to the *Leguminosae* family” [32]. Pulses’ protein contents range from the 18.4% of the Bambara bean to the 34.1% of the lupin. They not only contain a great amount of protein, but they also present the highest protein digestibility score among the plant-origin proteins. In addition, pulses are also a rich source of micronutrients such as iron, zinc and B-vitamins. Even though iron from plant origin is less absorbed by the human tract, when combined with meat, the absorption increases substantially [32]. Therefore, from a nutritional point of view, pulses are a great candidate as meat substitutes, providing high quantities of proteins and similar micronutrients to the ones in meat. Hence, several studies have analysed their role as meat extenders in the past fifteen years (Table 1). Even though soybean is not a pulse as per the definition, it is a legume, and for this reason, a study with texturised soy granules as a meat extender has also been included in Table 1.

Pulses in different forms have been used in meat product reformulations as binders or to increase their nutritional and healthier properties [40,41,42,43]. The starch, fibre and protein contents make pulses great binders, as they can form complex gel networks with meat proteins. These networks can trap the water and other compounds, forming stronger bonds between them and, thus, helping to achieve a higher retention in the meat matrix during processing [44]. Aslinah et al. [45] used adzuki bean flour as a fat and corn flour replacer in meatballs due to its water-holding capacities. Soy protein has been also widely used when developing reduced fat meat products due to its gelling properties [46,47]. The type and quantity of the pulse utilised, as well as the type of product, will determine the overall effect on the product stability in terms of WHC. In this regard, Nagamallika et al. [38] used two different pulses, Bengal gram flour and pea flour, to replace the meat content in chicken patties at two levels: 5% and 10%, yielding a higher stability at the higher level of substitution. Pea flour proved to yield a significantly lower cooking loss (9.5% vs. 30.3%) and higher emulsion stability (4.2% vs. 2.2%) and WHC (64.3% vs. 30.8%) when used at the 10% level compared to the Bengal gram flour patties. Nonetheless, at the 5% level of substitution, the emulsion stability and cooking loss were significantly lower for the patties with gram flour, whereas the WHC was significantly higher (19.8% vs. 47.8%). Yadav and Yadava [37] observed an increase in the yield and emulsion stability with an increasing level of substitution (3–9%) with gram flour in quail meat rolls. In a comprehensive analysis with 23 different type of pulses as meat substitutes in chicken and beef patties, this variation among the pulse types was also found [36]. The authors observed that cooking losses in beef patties ranged from 8.0% in the yellow split pea patty samples to 15.1% in the patties with pink beans, whereas the control had a 37.9% cooking loss. In the case of the pork patties, the control had a cooking loss of 22.9%, and the substitution improved the yield in all cases, with the cooking loss ranging from 5.6% to 10.7%, the lowest being the patties with small red beans and the highest for the ones with speckled butter beans. In an interesting study by Serdaroǧlu et al. [39], three different pulse flours (lentil, chickpea and blackeye bean) were used in low-fat meatballs, replacing not the meat but the rusk used in the control samples. Lower cooking losses and an increased WHC were found in the meatballs reformulated with pulses. This gives an idea that not only the starch (on higher quantities in the rusk) but the protein and fibre contents (much higher in the pulses) have big impacts on the water-holding capacities of meat products. The pulses with higher protein contents, blackeye beans and lentils, gave significantly higher yields to the meatballs. The substitution percentage also determined the effect of the extender on the product yield. Argel, Ranalli et al. [33] evaluated four different pulses (chickpea, lentil, green pea and bean) as extenders in pork patties with six different levels of meat contents. At the lower level of substitution (10.1%), the patties manufactured with bean flour had the highest cooking yields, followed by lentil and green pea, significantly different from the ones with chickpea flour. However, at the highest level of meat substitution (44.6%), the bean flour had the lowest cooking yield among the four pulses; the reformulated patties had higher cooking yields at all substitution levels than a commercial one. The different compositions of these flours might explain this, as the chickpea flour had the lowest protein and fibre contents but the highest fat levels. On the other hand, no significant yield changes were observed in dehydrated chicken ring meat using soy as a meat extender at a 5% level of substitution [34].

Another property closely related to the water-holding properties of meat products is the texture [48]. Texture is usually evaluated using a texturometer by means of a texture profile analysis (TPA) or a measurement of the hardness with the shear force value. A TPA analysis of pork patties substituted with pulse flour showed that the hardness and chewiness increased when compared to the control and commercial ones, but that this difference disappeared when the substitution level was above 35% and added water was at its highest level [33]. In the same study, the authors found that cohesiveness was lower in all the extended pork patties and that the bean flower had the lowest hardness among the four pulses studied. In addition to the level of substitution, the type of pulse will also affect the textural properties. An increase in hardness was observed when the rusk used in low-fat meatballs was replaced (10%) by the flour of three different pulses, being the meatballs with chickpea flour the ones with significantly higher hardness, followed by black bean and lentil flour [39]. Out of 23 varieties of pulses, only four pulses did not affect the shear force value when substituting 50% of the meat in beef patties and five pulses when pork patties were prepared instead [36]. The overall mean shear force was lower for the majority of the pulses used. As the substitution values for this study ranged between 35% and 50%, this agrees with the aforementioned results. The type of meat product will also affect how the substitution alters the textural parameters, as the networks formed in the matrix will be different depending on the degree of comminution and the quantity of fat, water and proteins. A great example can be observed in the 23-varieties study where the beef patties with green northern beans as the extender had the lowest hardness value, whereas in pork sausage patties, the hardness was one of the highest for this same pulse.

Colour is perhaps the attribute most difficult to mask when substituting meat with pulses, as not many of them have similar colour to meat. In addition, cooking of the meat product can also affect the colour changes generated by the use of pulses as extenders. Any colour comparison should be mainly addressed on the product at the state it is going to be purchased, although extra analyses can also be taken into consideration. Lightness was not affected by pulses as extenders in pork and beef patties with varying levels of substitution [33,36]. However, the use of some pulses as extenders in a variety of meat products have significantly affected the redness and yellowness values [33,34,36,39].

The product appearance is the first attribute the consumer observes before purchasing the product, and even though the instrumental colour measurements are correlated with the appearance, the consumer might not be able to detect the differences as the instrument does, or they could like better the colour change. In the same way, texture results from instrumental measurements and those from sensory panels differ substantially. Serdaroǧlu et al. [39] found that general appearance scores for meatballs with pulses as extenders did not significantly differ when compared to the rusk, but instrumental colour values for the same products showed significant changes in the yellowness value. In the same study, the meatballs with chickpea flour had the harder texture value, and it was scored lower by the panellists, but the one with the highest score was not the one with the softer texture but the second-hardest. These sensory analyses were done by trained panellists on a nine-point hedonic scale, and even though this practise should be avoided—hedonic analyses should always be carried out by non-trained panellists—it can give an idea of the sensorial properties of the product. When black beans, green grams and cowpeas were used as extenders in chicken seekh kababs, the sensory properties remained unaltered throughout storage, with no significant differences among the pulse varieties [35]. Yadav and Yadava [37] found that gram flour substituting meat in quail meat rolls at levels 3% and 6% did not affect the sensory properties, although, at 9%, they observed a significant decrease in the colour and flavour scores by the panellists. The use of texturised soy granules in dehydrated chicken meat only affected the meat flavour intensity, according to a sensory panel [34]. Argel et al. [33] found that pork patties where the meat was substituted (37%) with chickpea, lentil, green pea and bean flour emulsions had acceptable sensory properties, with no significant differences among the pulse types.

The use of pulses as meat extenders has been researched mainly in patties and similar meat products, but no work on comminuted ones, although some studies with pulses as binders can be found for these types of products [40,41,42]. In general, pulses seem to be an adequate ingredient to be used as a meat replacer, as they have a very similar nutritional composition and do not affect extremely the physicochemical properties of the finished product. Unfortunately, a limitation from pulses and legumes as extenders can be found on the allergenic potential of some proteins contained in soybean and peanuts that would restrict population access to these products (people with allergies) and would need proper labelling [49].

### 2.2. Other Meat Extenders of Plant Origin: Cereals, Tubers and Fruits

Other plants such as cereals, tubers and fruits have also been used in meat product formulations. The main reason of using these food products as ingredients in meat products has been the healthy properties they possess: high fibre contents, vitamins and minerals, important proportions of phytochemicals and antioxidants and void of cholesterol, among others [50,51]. Apart from their nutritional properties, some of these plants also have good functional and technological properties, such as improved water-binding and yield properties, fat emulsifiers, increased flavour, etc. [52]. Even though their main usage has been for the development of functional meat products [53,54,55,56], there has been also some research about the use of these ingredients as meat substitutes/extenders. Research about the use of cereals, tubers and fruits as meat extenders in the last thirteen years is summarised in Table 2.

Fruits and their by-products have been used as ingredients in meat products to improve the shelf lives and provide meat with antioxidants, fibre and other phytochemicals [66]. However, their role as meat extenders is yet to be explored, with only a few studies in the scientific literature. Melon flour, from defatted melon kernels, was used to substitute meat in beef sausages at levels 10–40% [63]. The authors found an increased yield, WHC and better sensory properties with the increasing levels of substitution. No significant differences with control on the overall acceptability and appearance were found at the 20% substitution level. The same authors found that, after two and four weeks of storage, the thiobarbituric acid reactive substance (TBARS) values were significantly lower for the sausages with substitution levels above 20% [64]. Low-fat beef patties where the meat was substituted with plum puree (5–15%) showed an increase in the cooking yield and redness of the patties but a decrease in WHC, lightness and yellowness [52]. The TBARS values of the extended patties with plum were lower at the end of the storage period, irrespective of the substitution level. In addition, the sensory properties were improved at the 10% and 15% levels of substitution, being the former the one with the best scores in overall acceptability, flavour, texture and juiciness. An increased cooking yield has been also found in beef patties extended with olive cake powder at levels 2.6–7.9% [59]. The olive cake powder also increased the amount of polyphenols and the antioxidant activity of the patties, but the instrumental colour was also affected, with a decrease of the lightness and an increase of the yellowness with increasing levels of substitution. The sensory properties were negatively affected, with significantly lower values at the higher levels of substitution. When using plum puree as an extender in beef patties (2.8–6.9% substitutions), the cooking yield and sensory attributes remained unaltered, but the WHC increased with the increasing levels, the redness dropped and the hardness increased [58]. All of these studies proved that fruits can be used as meat extenders, but further research is needed on different meat products (not only patties) and with different types of fruits and substitution levels.

Cereals are crops of the family *Gramineae*, which comprises nine species: corn, barley, millet, oat, rice, rye, sorghum, triticale and wheat. They are an important source of proteins (ranging from 7–18% dry matter) and vitamins (B group and E) [67]. Chicken patties where the meat was substituted by sorghum (5%), pressed rice (5%) and barley flour (10%) showed a significant decrease on the extract release volume and lower TBARS values at the end of storage, with no significant impact on the sensory properties [65]. Mishra et al. [34] found that rice flour at a 10% substitution level in dehydrated chicken ring meat did not affect the sensory properties, whereas a 5% meat substitution with barnyard millet flour decreased the sensory perception of the meat flavour intensity while not affecting any of the other sensory attributes. The same authors also observed that the yield was improved by these two extenders without affecting the instrumental colour. Both cereals also significantly reduced the cholesterol content and increased the manganese; the millet chicken meats had also a 10-fold increase of their iron contents, while the meats with rice had lower iron contents when compared to the control. Corn flour used as a meat extender in quail meat rolls increased the yield and emulsion stability with the increasing level of substitutions (3–9%) [37]. However, the sensory perceptions of colour and flavour were impacted on the rolls where meat was substituted at a 9% level but remained unaffected at the lower substitution levels. A screening of a combination of five different cereals and six plants and tubers as meat extenders (10%) in sheep meat cubes was performed using a Plackett-Burman design [62]. The authors found that millets, carrots and cabbages gave the cubes the most desirable sensory characteristics and that further research with these ingredients should be guaranteed. Malav et al. [61] analysed the use of a blend of sorghum with potato, lentil and water chestnut flours as extenders (15%) in restructured chicken. The blend of extenders exerted higher yields and similar texture attributes but lower sensory scores. Another study where cereals were combined with other ingredients as meat extenders in the same type of product was done by Gupta and Sharma [57]. Wheat, oat and barley were blended with potato, whey and texturised soy protein in three different combinations that were compared to a control. The three blends increased the cooking yield and decreased the hardness, but only one of them did not differ in the overall acceptability of the product; the other two had lower scores for flavour. With regards to the instrumental colour, the redness was not affected, but the yellowness increased in all the reformulated samples. However, the sensory appearance was higher for the sample with the highest chroma value. Cereals proved to be important and successful ingredients when used as meat extenders, but their behaviours in meat products different than restructured meat and chicken are still unknown. It is also important to highlight that cereals containing gluten (wheat, rye, barley and oats) have allergenic potentials that must be declared in the labelling.

### 2.3. By-Products of the Food Industry as Meat Extenders

The food industry (from vegetables or animal products) produces high amounts of residues and/or by-products that are edible compounds with high percentages of proteins and/or fibres. In today’s global scenario, the use of these compounds—in many cases, undervalued—could be an opportunity to replace meat for manufacturing more sustainable meat products [68]. Furthermore, many of these residues are a source of polyphenols, organic acids and fatty acids, among others, which are underutilised, providing added value to the products in which they are included [69]. In this regard, some studies have assayed the use of residues from the agri-food industry as meat extenders (Table 3).

Okara is a by-product with low commercial value that is generated in massive volumes (about two to three tons for each ton of soybean processed) during the manufacturing of soymilk and tofu [80]. This component presents solvent-binding properties, making it an ideal low-cost ingredient to increase yields in meat products (Table 3). Moreover, okara contains valuable components such as fibre and high-quality protein (40% on a dry weight basis) due to the presence of a good essential amino acids profile and its digestibility [80]. In that sense, okara has been applied to extend meat contents both in fresh and cooked emulsion-based meat products (Table 3). In beef burgers, lean meat has been replaced by wet okara in different quantities, up to 37.5% (Table 3). In general, it was observed an increase of the moisture content and a decrease of the protein level in the reformulated burgers [70,71,73]. Moreover, Tie Su et al. [73] obtained beef burgers with 60% less calories than commercial products when 12% of okara was added. The use of okara as a meat extender improved the cooking yields of the samples [70]. Tie Su et al. [73] noticed that, as the percentage of okara increases, an increase in hardness occurs, while Strada de Oliveira et al. [71] observed an improvement in tenderness with respect to the control samples. The effect of wet okara on the sensory properties was significant, and higher scores for overall acceptability were recorded for products with approximately 20% added okara [70,73]. In cooked emulsion-based sausages, contrary to those observed in fresh meat products, the moisture content was increased with an okara addition [72]. Water and oil-holding capacities were improved as a consequence of okara additions, and in that sense, the cooking yield was improved [72]. For textural properties in cooked emulsion-based products, the incorporation of okara presented contradictory behaviours. The same authors observed an increase for the texture parameters with up to 40% of okara added to beef sausages, while a decrease of the hardness, chewiness and breaking force occurred when okara was incorporated in pork meat batters [74]. On the other hand, the overall acceptability of the samples decreased with the okara incorporation [72].

The residue obtained from the production of cashew apple juice (skin and the husk) has been used to extend beef meat in the formulation of hamburgers. With increasing the concentrations of the residues, the samples showed lower moisture, protein and lipid levels, while their fibre contents were higher. Hamburgers with improved yields and similar flavours than the control were observed with additions up to 10.70% of the residue [75]. Apple pomace powder was employed (2–8%) to replace buffalo meat in emulsion-based sausages by increasing the fibre contents. Moreover, the cooking yield and emulsion stability got enhanced [76].

Mushroom by-products are described as a good source of protein, dietary fibre and phenolic components, with the potential to be strong antioxidants [77]. In that sense, the use of different amounts (2%, 4% and 6%) of enoki (*Flammulina velutipes*) mushroom stem wastes as meat extenders in nuggets enhanced their composition (Table 3). The inclusion of meat extenders improved the oxidative stability and shelf-lives of treated nuggets without impacting the sensory attributes of reformulated nuggets.

Whey is a by-product of the dairy industry, which is generated in massive quantities during the manufacture of cheeses, yogurts and other dairy products [81]. Its great content of high biological value proteins offers interesting possibilities to be used during the processing and manufacturing of meat products. Hale et al. [78] extruded a dry whey protein concentrate (80% protein) to obtain an ingredient that they used to substitute from 0% to 50% of beef in the elaboration of patties. Samples containing up to 40% of whey extrudes were as acceptable to a consumer panel as all-beef patties. Moreover, the cooking yield was improved, and these patties suffered less diameter reductions and less water and fat losses by the cooking process.

The meat industry also generates compounds that hold strong potential for higher-value techno-functional applications due to their high-quality protein contents (Table 3). However, their use as meat extenders in meat products is very limited. For example, Álvarez et al. [79] extracted protein concentrates from different residues of the meat industry to be used as meat replacers in the elaboration of an Irish breakfast-type sausage: blood plasma, exudates generated from ham elaboration, brine solutions and water produced during edible fat processing. Two levels were assayed: 15% and 30% (Table 3). Regarding the composition, raw products showed lower fat contents and higher protein levels than the control ones. However, the technological properties were conditioned by the type of protein used and the level of meat substitution. In general, for all types of protein, the 15% meat replacement offered products with a better overall final product quality. Regarding the type of protein, plasma proteins at both replacement levels had the most positive effect on the technological properties, whereas the use of protein concentrates from brine solutions to substitute meat resulted in sausages with lower fat and water-binding properties and redness values (Table 3).

Based on the foregoing, it seems that the use of residues or by-products from the food industry as meat replacers endows products with compounds that offer positive effects on health without being a detriment to their technological properties. In addition, this strategy offers multiple advantages to maintaining a more sustainable world by both using industrial residues and reducing meat productions.

### 2.4. Mushrooms as Meat Extenders

Fungi have been used in human foods for a long time. Of more than 14.000 species of mushrooms, at least 2000 of them have various degrees of edibility [82]. Mycoprotein is fungal in origin, and it is utilised as a high-protein, low-fat, health-promoting food ingredient [49]. Mycoproteins could be obtained by the continuous-flow fermentation of *F. venenatum* on a glucose substrate, and it is used to elaborate meat analogues. However, in the development of more sustainable meat products, some studies were carried out adding mushrooms directly to meat products (Table 4), replacing different proportions of meat proteins by mycoproteins.

Mushrooms are a good source of dietary fibre, where approximately one-third is chitin and two-thirds β-1,3 glucan and 1,6 glucan. Chitin is a modified polysaccharide with an analogous structure to cellulose and considered an insoluble fibre with potential prebiotic properties in gut microbiota [89]. In addition, mushrooms are also a source of proteins; essential amino acids; vitamins (such as thiamine, riboflavin and niacin) and essential minerals (such as Ca, P, Mg, Cu, Se and Zn). Moreover, these products are low in calories, fat and sodium [90]. In that sense, the application of mushrooms as meat extenders could also be an opportunity to improve the presence of health-promoting bioactive components in meat products.

White mushrooms (*Agaricus bisporus*), the most cultivated edible mushroom, poses a dual opportunity as a meat extender by reducing the meat content while also imparting flavours that can complemented and enhance the saltiness perception [82]. Wong et al. [83] compared two meat extenders, a traditional one (textured soy) and *Agaricus bisporus*, to replace 10% to 50% of meat in the development of beef patties (Table 4). Increasing textured soy improved the cooking yield of patties but did not affect their colour or textural properties. However, increasing the level of mushroom extenders performed statistically similar to an all-meat control in yield, lightness and redness, while decreasing the mechanical properties. Additionally, meat extension using mushrooms yielded sensory liking scores more similar to the all-meat formulations than textured soy in reduced sodium samples. In the same way, white mushrooms were used to replace meat in two meat-based dishes, carne asada and beef taco blends, whose sodium contents were reduced [84]. In carne asada, the beef substitution (50%) with mushrooms did not alter the overall flavour strength of the dish, but the replacement of 50% or 80% of meat in the beef taco blend enhanced its overall flavour. The ability of mushrooms to mitigate sodium reductions in terms of the overall flavour has been attributed to the fact that mushrooms contain umami tastants [82]. White jelly mushroom (*Tremella fuciformis*) is another type of edible mushroom that has been used as a meat extender in pork meat patties (Table 4) [85]. In this case, higher mushroom quantities (30%) decreased the sensory acceptance of patties because of the mushroom flavour. However, patties containing 10% of mushrooms improved significantly the sensory affections due to their oil-holding capacities. Furthermore, this ability, along with its capacity to bind water, allowed improving the cooking yield of patties formulated with white jelly mushroom [85]. In pork sausages, *Lentinula edodes* has been used as meat extender to replace 25%, 50% and 100% of the meat (Table 4). Regarding sensory acceptability, all samples were satisfactory. Although those with 25% of substitutions showed the highest scores for sensory attributes. From a technological point of view, the presence of mushrooms improves the oxidation stability and the cooking yield of sausages [86].

The use of *Pleurotus sajor-caju* as a meat extender (25% and 50% of meat substitutions) in beef patties and in lower proportions (2% to 6%) to replace chicken meat in the formulation of frankfurters produced an increase of their fibre contents. It should be noted that this fibre was insoluble mainly based on β-glucans (0.78 g/100 g in the case of patties and 1.43 g/100 g in frankfurters) [87,88]. As with other mushrooms, the use of *Pleurotus sajor-caju* as a meat replacer improved the cooking yield of the products. The hardness values of the reformulated products were lower. However, the sensory analysis scores indicated that the products were accepted by the panellists [87,88].

Mushrooms seem to be an adequate ingredient to be utilised as a meat replacer. The use of mushrooms allows for the development of healthier meat products with higher fibre and less salt contents (as they have the potential to increase saltiness perceptions) without affecting much the physicochemical properties.

### 2.5. Insects as Meat Extenders

Entomophagy, or the practise of eating insects, is a long-time practise and an important nutritional source (high-quality protein, lipids, carbohydrates, mineral elements and certain vitamins) for many cultures, mainly located in Africa, Asia and Latin America [91]. More than 40 years ago, Meyer-Rochow [92] already suggested that insects could supplement traditional animal protein sources. Currently, there is a growing interest in edible insects as a novel source of protein due to their high contents, as well as their functionalities, which have been described similar to conventional proteins (included meat proteins) [91]. However, probably due to insect food neophobia in Western countries, there are only a few studies using insects as meat extenders, and the majority are from Eastern Asian countries (Table 5). With the aim to decrease this well-known food neophobia related to insects, Caparros Megido et al. [93] decided to test the level of sensory-liking of patties in which beef was replaced (53%) by mealworms, allowing them to hide insects and to present them in a familiar way. The authors concluded that insect integration into Western food culture could be feasible, as the taste and appearance of burgers were rated higher than neutral scores, positioning them between a fully meat burger and a fully vegetable burger.

The incorporation of mealworms as meat replacers was also studied to evaluate their effects in the composition and technological properties of new products. Ju-Hye et al. [94] studied the effects of different replacement ratios (10% to 60%) of pork meat in the development of patties (Table 5). The addition of mealworms conditioned significantly the composition of the samples, decreased protein contents and increased fat levels. The cooking yield was improved with the presence of insects. There were no significant differences in the sensory characteristics of burgers, although the shear force was reduced and the lightness was increased with the replacement of meat by insects.

In emulsion-based meat products, mealworms (*Tenebrio molitor* L.) have been used to replace 10–60% of pork meat (Table 5). Reformulated samples had increased protein and fat contents when the meat was replaced at the 10% level [95,96]. However, Choi, Kim, Choi, Park, Sung, Jeon, Paik and Kim [95], who assayed higher levels of extended meat (up to 60%), observed that frankfurters with a higher meat replacement by mealworms increased the protein content but decreased the fat content approximately to 30% in respect to all-pork meat samples. Moreover, the incorporation of edible insects increased the mineral contents of emulsion sausages [96]. The cooking yield was improved with a substitution of meat of 10%; extended higher meat decreased the cooking yield [95,96]. Additionally, replacing pork meat with up to 10% mealworms successfully maintained the sensory quality of frankfurters.

Silkworm pupae (*Bombyx mori*) and the House cricket (*Acheta domesticus*) are two other types of edible insects used as meat extenders (Table 5). Kim, Setyabrata, Lee, Jones and Kim [96] added freeze-dried Silkworm pupae (*Bombyx mori*) to replace 10% of the pork meat in an emulsion-based meat product. They assayed three strategies to incorporate the insects: ground, defatted and acid-hydrolysed. The inclusion of insects had no impact on the protein solubility of emulsion sausages. The protein contents of sausages were increased for all the treatments; however, the fat contents only were increased when insects were ground. Additionally, the mineral content was improved when ground and defatted Silkworm pupae was incorporated [96]. The replacement of pork meat with house cricket flour within a 10% level could fortify the product with proteins and some micronutrients (phosphorus, potassium and magnesium) without a negative impact on the cooking yield and textural behaviours [97].

Edible insects possess the necessary physical properties to be used as an alternative nonmeat ingredient for incorporation within fresh or emulsified meat products, which could be further promoting to improve the image that the consumers have of them. Moreover, the addition of invisible insects in food preparations helps to reduce insect food neophobia [93].

## 3. Meat Products′ Sustainability from the Consumer Perspective

As stated before, protein production has a large impact on the climate change, with proteins from meat being much less sustainable than plant-based proteins [98]. It seems logical to think that the daily choice of food has a high impact on the environment, and therefore, acting to change consumer preferences seems an appropriate strategy to reduce the negative impact that food production may have [99].

Some alternatives for meat products made entirely of vegetable components (e.g., tofu) can be already found in the supermarkets, although the market shares of these products are still very low compared to meat and meat products. The lower penetration of these products in households could be partially explained by the lack of texture and taste reported for some of them [100]. In addition, the heavy processing conditions to obtain these products and, in consequence, the multiple additives that they contain are sometimes neglected; besides, they can have a really high carbon footprint [28].

Complex external cues (perceived healthfulness, animal welfare, environmental impact and sustainability) are increasingly taken into account in our preference for meat [101]. However, despite a seemingly close match between the consumers’ image of a sustainable, healthy and a plant-based diet [102], there is actually low consumer awareness of the environmental impact of meat production, as well as a low willingness to change meat consumption behaviours in terms of reducing or substituting meat in Europe. It is therefore relevant to determine the opportunities and barriers for consumers to adopt such alternative meat protein sources in their diets [100]. Preconceptions towards vegetarian diets, habits and prices and a lack of familiarity with meat substitutes, among others, are barriers to changing meat consumption behaviours [103]. Despite all of the above, it must be taken into account that the complete elimination of meat from our diet is impractical and might even have negative societal consequences [104].

The challenge of developing healthier foods with high consumer appeal underscores the need for integrated culinary, sensory and consumer research in this area [105]. Although Hoek et al. [99] concluded that, for the development of new foods, more emphasis is needed on consumer evaluation instead of on the sensory properties of the individual product. In that regard, studies that also take consumer behaviours into consideration could be an alternative to standard consumer sensory analyses. A recent alternative method called Mind Genomics has been applied on meat analogues, with promising results [106]. In addition, in order to increase the acceptance of novel products, it is necessary to obtain knowledge about the demographics, the consumption patterns and the sensory drivers of consumers [107]. In Western countries, vegetable proteins have a high level of acceptance and are consumed regularly. However, the same does not occur with the inclusion of nonconventional meats, insects or food by-products in our diet.

An alternative to conventional meat production is the use of more sustainable species like rats or other pest rodents [108,109]. Although rats are a regular staple in some Asian regions, the mere suggestion of its consumption in Western countries generates a big consumer rejection. Caparros Megido et al. [93] concluded that insect-tasting sessions are important to decrease their neophobia, because they observed that people with previous entomophagy experience gave globally higher ratings to meat products that contained insects-based proteins. In addition, Meyer-Rochow and Hakko [110] concluded that the acceptability of insect consumption would be higher if they were presented in flours or pastes. The inclusion of food by-products or residues from the meat industry can also present a challenge to consumer acceptance. Even though this practise presents a double opportunity towards healthier and more sustainable meat products, their acceptance is quite limited. Some of the reasons are related with consumer perceptions of these by-products as actual waste and, thus, unhealthy, but even if healthiness would be proven, consumers would also reject some of these reformulated products due to “ideational” reasons [111]. This concept is linked to the sensation of disgust some products produce in consumers just because of their origin (e.g., insects, by-products, etc.) and bad taste.

Meat eating is a habitual behaviour that is difficult to change; there is an unwillingness to reduce or substitute meat among the vast majority of consumers in various European countries [100]. In search of new alternatives, it is necessary to know how different food-related attitudes and behaviours (food choice motives, food fussiness, etc.) and socio-demographics (gender, age, education, etc.) influence the consumption of such protein sources [103]. In that sense, although some studies concluded that there is an urgent need for meat moderation campaigns that provide a broad spectrum of measures and habit-breaking interventions—including the promotion of vegetarian options [112]—the use of extenders to reduce animal proteins in the development of meat products could help to minimise their environmental impact without having to give up entirely the meat products in our diet.

## 4. Conclusions

A global demand for high-protein foods is on the rise. Meat and meat products are an important protein source in our diets but also great contributors to environment degradation through the far-from-sustainable production and increased carbon footprint of the finished products. Alternatives to more sustainable protein productions fall into two categories: mitigation of the negative impact and the use of more sustainable protein sources. With the use of meat extenders in meat products, we would be mitigating their negative impact by reducing the meat content, but we would also be maintaining the nutritional properties (i.e., protein and minerals) by using more sustainable sources. Even though pulses are the main extenders we should be looking at—similar nutritional profiles to meat—there are other extenders worth exploring. Apart from mushrooms, cereals, tubers and fruits that can be a great choice for some types of meat products, novel approaches such as insects and by-products from the food industry present an opportunity to develop healthier and more sustainable meat products. However, there is a need to devise strategies to increase consumer awareness and acceptance of these types of products. The plethora of sources and possibilities make the use of meat extenders the most viable and interesting approach towards the production of more sustainable meat products.

## Figures and Tables

**Table 1 foods-09-01044-t001:** Use of pulses as meat substitutes (extenders).

Ingredient Used	Meat Product	Meat Substitution (%)	Effect on Properties	References
Green pea flour emulsion	Pork patties	10.1–44.6	Higher yield, lower redness values, increased yellowness at higher substitution levels, increased hardness at all levels	[33]
Chickpea flour emulsion	Pork patties	10.1–44.6	Higher yield, lower redness values, increased yellowness at higher substitution levels, increased hardness at lower substitution levels	[33]
Lentil flour emulsion	Pork patties	10.1–44.6	Higher yield, increased hardness at lower substitution levels	[33]
Bean flour emulsion	Pork patties	10.1–44.6	Higher yield, lower redness values, increased hardness at lower substitution levels	[33]
Texturised soy granules	Dehydrated chicken ring meat	5	Lower meat flavour, lower yellowness hue and chroma	[34]
Cowpea	Chicken seekh kababs	15	Sensory properties not affected, lower TBARS * values, higher microbial counts	[35]
Green gram	Chicken seekh kababs	15	Sensory properties not affected, lower TBARS values, higher microbial counts	[35]
Black bean	Chicken seekh kababs	10	Sensory properties not affected, higher microbial counts	[35]
23 different pulses	Beef patties	35–50	Higher yield (highest for yellow split bean), colour values not affected with most of pulses, texture not different from control on black-eyed pea, baby lima bean, purple hull pea and crowder pea patties	[36]
23 different pulses	Pork sausage patties	35–50	Higher yield (highest for small red), colour values not affected with most of pulses, texture not different from control on black bean, lentil, black-eyed pea, green split pea and baby lima bean	[36]
Bengal gram flour	Quail meat rolls	3–9	Higher yield, sensory not affected at 3–6%, lower protein	[37]
Bengal gram flour/Pea flour	Chicken patties	5–10	Pea flour higher yield, stability and sensory scores than gram flour at higher levels of substitution	[38]
Blackeye bean flour	Meatballs	10	Higher yield and overall palatability, lower yellowness and tougher compared with Rusk	[39]
Chickpea flour	Meatballs	10	Higher yield, lower yellowness and tougher compared with Rusk	[39]
Lentil flour	Meatballs	10	Higher yield, lightness and overall palatability, lower yellowness and tougher compared with Rusk	[39]

* TBARS: thiobarbituric acid reactive substances.

**Table 2 foods-09-01044-t002:** Use of cereals, tubers and fruits as meat substitutes (extenders).

Ingredient(s) Used	Meat Product	Meat Substitution (%)	Effect on Properties	References
Blend of potato, soy protein, oat meal, barley flour, whey protein concentrate	Restructured spent hen	23.5–25.5	Sensory properties not affected and higher yield. Softer texture and increased chroma values	[57]
Dried pumpkin pulp and seed	Beef patties	2.8–6.9	Increased water-holding capacity (WHC), lower redness, no changes in texture and sensory	[58]
Olive cake powder	Beef patties	2.6–7.9	Lower sensory scores, higher protein and yield, increased yellowness	[59]
Different blends of fibre, carrageenan and pork rind	Beef and chicken sausage	35–50	Decreased hardness, similar flavour to control but loss of general sensory quality, with the exception of a few blends	[60]
Rice flour	Dehydrated chicken ring meat	10	Sensory properties not affected, higher yield, lower iron	[34]
Barnyard millet flour	Dehydrated chicken ring meat	10	Higher yield, multiplied iron content, lower meat flavour	[34]
Blend of lentil flour, sorghum, potato and water chestnut flour	Restructured chicken meat blocks	15	Higher yield, similar texture properties, lower sensory scores	[61]
Plum puree	Beef patties	5.1–15.4	10% substitution best sensory results with no detrimental effects on physicochemical properties	[52]
Corn flour	Quail meat rolls	3–9	Higher yield, sensory not affected at 6%, lower protein	[37]
Several cereals, tubers and plants	Meat cubes	10	Pearl millet, carrot and cabbage showed highest-ranking scores in sensory properties	[62]
Melon flour from kernels	Beef sausages	10–40	Higher yield, no changes in sensory attributes up to 20% substitution. Lower TBARS values.	[63,64]
Sorghum flour	Chicken patties	5	Lower TBARS at end of storage, sensory properties not significantly different	[65]
Barley flour	Chicken patties	10	Lower TBARS at end of storage, sensory properties not significantly different	[65]
Pressed rice flour	Chicken patties	5	Lower TBARS at end of storage, sensory properties not significantly different	[65]

**Table 3 foods-09-01044-t003:** Use of by-products of the food industry as meat substitutes (extenders).

Ingredient Used	Meat Product	Meat Substitution (%)	Effect on Properties	References
Okara	Beef patties	7.5–37.5	Cholesterol reduces for raw (6–56%) and cooked (9–42%). Higher cooking yield, pH, lightness and yellowness. Sensory attributes valued negatively with 37.5% of meat replacements.	[70]
Okara	Beef burger	5–25	Increase lipid and moisture contents. Higher luminosity and dimmed during storage. Changes in the brown colour	[71]
Okara	Beef sausages	10–40	Carbohydrate, ash and fibre contents increased, while moisture, fat and protein contents decreased. Improved WHC but decreased textural parameters	[72]
Okara	Beef burgers	6 and 12	Sixty percent less calories. Increased hardness but decreased cohesiveness, chewiness and springiness. Lower sensory scores with 12% of substitutions	[73]
Okara	Pork meat gels	3–27	Higher cooking yield. Increased in lightness, hardness, chewiness and breaking force of gels but decreased in cohesiveness. Higher storage (G′) and loss (G”) modulus by heating.	[74]
Cashew apple residue powder	Hamburgers	7.1–14.3	Reduced 35% of the lipid content and increased of up to 7.6% of the fibre. Lower humidity but sensorial acceptable with 7.1 and 10.7% of meat replacements	[75]
Apple pomace	Buffalo emulsion-based sausage	2–8	Increased fibre content and improve cooking yield and emulsion stability	[76]
Enoki (*Flammulina velutipes*) mushroom stem waste powder	Goat nuggets	2–6	Increased dietary fibre, ash and phenolics compounds. Improved the emulsion stability, WHC, oxidative stability and shelf-life. Slight hardness decrease. No negative effects in the sensory attributes.	[77]
Textured whey proteins (TWP)	Beef Patties	0–50	Higher cooking yields. Patties containing up to 40% of hydrated TWP obtained similar sensory evaluations than all-beef patties	[78]
Protein concentrates from porcine blood	Irish breakfast sausage	15 and 30	Higher protein contents in raw samples. Decreased fat levels in cooked samples. Higher cooking yield and WHC for 15% of replacements. Decreased hardness and chewiness with 30% of meat substitutions	[79]
Protein concentrates from pork hams exudates	Irish breakfast sausage	15 and 30	Lower fat contents in raw samples. Higher protein contents with 30% of meat replacements. Decreased WHC. Decreased hardness and chewiness values with 30% of replacements	[79]
Protein concentrate from residues of edible fat processing	Irish breakfast sausage	15 and 30	Decreased fat contents. Similar WHC and cooking yield. Decrease hardness and chewiness values with 30% of meat replacements	[79]
Protein concentrate from brine solutions	Irish breakfast sausage	15 and 30	Higher protein contents. Higher cooking losses. Decreased redness in raw samples but increased when they are cooked	[79]

**Table 4 foods-09-01044-t004:** Use of mushrooms as meat substitutes (extenders).

Ingredient Used	Meat Product	Meat Substitution (%)	Effect on Properties	References
Mushroom (*Agaricus bisporus*)	Beef Patties	10–50	Allows reduced sodium patties (1.5% NaCl). Increasing mushroom extender level; samples perform similar to an all-meat control in yield, lightness and redness; increase moisture and yellowness and decrease mechanical properties, sodium and fat contents.	[83]
White mushroom (*Agaricus bisporus*)	Beef taco blend	50 and 80	Enhancement of overall flavour and mitigated salt reduction.	[84]
White mushroom (*Agaricus bisporus*)	Carne Asada	50	Allows reduced sodium samples. No alterations on the overall flavour strength.	[84]
White jelly mushroom (*Tremella fuciformis*)	Pork Patties	10–30	Improve cooking yield and increase lightness and yellowness. Ten percent substitution improved the sensory acceptance, while 30% decreased the approval of patties.	[85]
*Lentinula edodes*	Pork sausage	25–100	Increased moisture, fibre, essential amino acids and total phenolic content. Higher cooking yield and antioxidant activity. Decreased protein, energy ash, pH and texture parameters. Twenty-five percent substitutions are the best sensory acceptance.	[86]
*Pleurotus sajor-caju*	Beef patties	25 and 50	Increased insoluble fibre content, mainly β-glucan. Decreased fat retention during the cooking process. Best cooking yield with 25% of substitutions. No differences in sensory attributes.	[87]
*Pleurotus sajorcaju*	Chicken frankfurters	2–6	Decreased fat content. Enhancement of dietary fibres up to 6.20% and β-glucan up to 14.30%. Hardness was decreased.	[88]

**Table 5 foods-09-01044-t005:** Use of insects as meat substitutes (extenders).

Ingredient Used	Meat Product	Meat Substitution (%)	Effect on Properties	References
Mealworm (*Tenebrio molitor* L.)	Pork patties	10–60	Improved cooking yield. Higher fat content. Decreased moisture and protein content. Lower lightness but higher force shear. No sensory characteristics affected	[94]
Mealworm larvae (*Tenebrio molitor*)	Burger patties	53	The appearance of insect-based burgers was preferred by men. In terms of overall liking, meat substitution by insects was better valuated than by legumes	[93]
Mealworm larvae (*Tenebrio molitor* L.)	Frankfurter	10–60	Decreased moisture and fat content while increased protein level. Decreased lightness and textural parameters. Greater replacement than 15%. decreased emulsion stability. Less sensory acceptance	[95]
Mealworm larvae (*Tenebrio molitor*)	Emulsion sausage	10	Increased protein and mineral contents but decreased moisture. Improved cooking yield. Samples with more lightness but with lower values for textural parameters.	[96]
Silkworm pupae (*Bombyx mori*)	Emulsion sausage	10	Increased protein and mineral contents but decreased moisture. Improved cooking yield. Samples more lightness but with lower values for textural parameters	[96]
House Cricket (*Acheta domesticus*)	Emulsion sausage	5 and 10	Increased protein and minerals (P, K and Mg), no negative impacts on cooking yield and textural properties	[97]

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
