# Peer review of "Towards More Sustainable Meat Products: Extenders as a Way of Reducing Meat Content"

_foods, 2020, doi:10.3390/foods9081044_

Round 1
Reviewer 1 Report
Ref Report 303
Full Title: Towards More Sustainable Meat Products: Extenders as a Way of Reducing Meat Content
by Tatiana Pintado and Gonzalo Delgado-Pando
This manuscript by T. Pintado and G. Delgado-Pando is in many places excellent, but seemingly biased and even wrong in some. However, the authors can easily revise the criticsed sections and turn this ms into an important contribution!
On page 2 the authors claim that “people following vegan and vegetarian diets [are] in need of supplementation to achieve DRI for this micronutrient”. Although correct for a vegan diet, it is incorrect for vegetarians, because a vegetarian diet contains milk and milk products like yoghurt and cheese. T least 500 million Indians live a completely vegetarian and healthy life!
On page 13 the authors claim that “Meat eating is a habitual behaviour that is difficult to change…”, which is wrong. Human ancestors were frugivores/vegetarians (the great apes still are) and the human digestive system with carbohydrate-attacking enzymes in the saliva, lack of canine teeth, the long human intestines plus caecum, colour vision (carnivores lack colour vision), etc. all suggest that human habitual behaviour was largely meatless. Moreover, there is no unwillingness to reduce or substitute meat, for even in countries with large numbers of meat consumers more and more people turn to a meatless and vegetarian diet (as noticeable, e.g. in Scandinavian schools that always have meatless dishes on their menu and often even have one totally meatless day per week for everyone or in Japan, where more and more people turn to a vegetarian diet).
The manuscript before me, of course, deals with meat “extenders”, which is an excellent approach in the context of the problems associated with animal protein production. However, maybe the authors could also have, at least briefly, mentioned alternatives to the current conventional meat producing species, namely small rodents, e.g. rats (cf. Journal of Ethnobiology& Ethnomedicine), etc. which as protein sources would have much less environment impact that currently used domestic species. The authors need to consider that not all saturated fatty acids are bad; in fact some have been credited with superior properties to unsaturated fatty acids, e.g., Mensink R.P. 1993 Effect of the individual saturated fatty acids on serum lipids and lipoprotein concentrations. Am J. Clin. Nutr. 57 (5) Suppl. 711S-714S
Bonanome A. and Grundy, S.M. 1988 Effect of dietary stearic aid on plasma cholesterol and lipoprotein levels. New England J. Med. 318: 1244-1248
Although generally quite well written, I encourage the authors to seek help from someone with an excellent command of English to go over the text once more as there are places in which grammar and style need to be corrected. Some additional comments of mine I wish the authors to consider:
Line 29: The date 2025 is wrong. Most likely it should be 2015.
Line 34: Authors should continue the sentence “….demand for animal protein, which was already addressed by Meyer-Rochow (1975) more than 40 years ago, suggesting that insects could supplement traditional animal protein sources [1, 2]. Full reference: Meyer-Rochow, V.B. Can insects help to ease the problem of world food shortage? Search 1975, 6, 261-262.
Line 43: contribute (not ‘contributes)
Chapter 2: Meat Extenders and section 2.1.: excellent; 2.2.: excellent; 2.3: excellent.
Line 311 and section 2.4.: I think the authors ought to mention (I know they doin one sentence) that mushrooms contain chitin, which is largely indigestible for humans, with the exception of a small percentage of chitinase-possessing humans (e.g. “Human Gastric Juice Contains Chitinase That Can Degrade Chitin”. Ann Nutr Metab 2007;51:244–251 by Maurizio G. Paoletti, Lorenzo Norberto, Roberta Damini, Salvatore Musumeci).
In the setion 2.5. Insects…
Line 402: what do the authors mean by Easter countries? Easter is a Christian holiday! Either in this section or in 3. ….consumer perspective, quoting a study by Meyer-Rochow, V.B. and Hakko, H. 2018 (“ Can edible grasshoppers and silkworm pupae be tasted by humans when prevented to see and smell these insects? Journal of Asia-Pacific Entomology 21 (2018) 616–619) might be in order.
This can become a really important and interesting paper!
Reviewer 2 Report
In this review, the authors aimed to give an overview of the compounds that can be used as meat extenders, and how they can affect the physicochemical and sensory properties of reformulated products. The manuscript is so organized and very well-written. However, I have some minor comments.
- I recommend reviewing the manuscript for some minor grammar/punctuation errors.
- The introduction is well written but I recommend keeping it shorter.
- To give an example regarding the environmental impacts of replacing 25–50% of animal-derived foods with plant-based foods, I recommend citing the paper by Westhoek et al. (https://doi.org/10.1016/j.gloenvcha.2014.02.004). I also recommend citing a recently published paper by Vatanparast et al. (https://doi.org/10.3390/nu12072034) which is regarding the impact of decreasing red and processed meat and increasing plant-based meat alternatives on the diet quality and nutrient intakes.
- Page 2, line 54, please define DRI in the text.
- Page 2, lines 65-57, please provide a reference for this sentence.
- Page 2, line 71, “the reduction of components to appropriate amounts,” reduction of which components?
Page 2, line 72, “The former is focused on the reduction saturated fat,” it seems (of) is missing in the sentence. - Page 2, line 77, please change “alternative” to “alternatives.”
- Page 3, Table 1, please define “TBARS” in the table footnote.
- Page 4, lines 151-153, please provide a reference for this sentence.
- Page 5, lines 168-170, please provide a reference for this sentence.
- Page 5, lines 194-196, this sentence needs revision.
- Page 5, line 198, “although some studies with pulses as binders can be found for this type of products,” please provide references.
- Page 5, lines 197-201, please mention the allergenic nature of some legume crops such as soybeans and peanuts as a limitation.
- Page 5, lines 204-206, this sentence needs revision.
- Page 6, line 219, please define WHC. Please be consistent about using WHC or water holding capacity in the text.
- Page 8, lines 281-282, please provide a reference for this sentence.
- Page 9, line 300, “with up 40% of okara” do you mean “with up to 40%”
- Page 9, line 308, “addition up 10.70%” please revise.
- Page 12, line 421, please change “decrease” to “decreased.”
- Page 12, line 434, “without negatively impact on cooking” please change negatively to negative.
- Page 13, line 450, “the multiple additives that they content,” do you mean that they contain?
- Page 13, lines 466-468, this sentence needs revision.
Reviewer 3 Report
General comments:
The paper titled ‘Towards More Sustainable Meat Products: Extenders as a Way of Reducing Meat Content’ focuses on the problem of sustainability in meat products. This is a very up-to-date and interesting topic. The review provides a broad and comprehensive description of all the issues related to non-meat substances used to partially replace meat in meat products. All the processes and terms are explained and presented in the paper sections in a clear and well-structure manner.
There are a few things lacking, that have to be checked by the Authors – see detailed comments.
Detailed comments
Line 270. ‘By-products/waste of food industry as meat extender’ The term ‘waste’ suggests an incorrect definition of edible by-products.’ Waste’ is associated with something that cannot be re-used. Edible by-products are not the main product and differ from the main product, but they still are valuable. Using the term waste suggests something totally opposite.
Table 3.Though animal by-products are mentioned, they are described only partially. Why? Porcine blood or fat are not the only by-products used. What about internal organs? Examples of internal organs used should also be added.
Author Response
We would like to thank Reviewer 3 for the time and effort contributing to improve the manuscript. Please see attached a pdf document with our response to the queries from reviewer 3.

Reviewer 4 Report
L29: 2025?
L44-45: „…meat and meat products should not be disregarded in the diet as they…” The authors aimed to write a review about meat extenders. Please rephrase.
L49-50: Rippin et 50 al. [7] – check reference style
L51-52: There is no unique food alternative to meat or meat products with similar nutritional profile and even a combination of several foods does not assure the same nutritional intake. – what about vegetarians, vegans? Please rephrase
L53: animal origin
L54: define DRI
L59: reference missing
L64: use such as instead of like
L68: define SDG
L68: reference is missing
L72: reduction of
Introduction. There are only two points explained about why it would be important to reduce meat consumption: sustainability and health issues. However, it should be noted that meat substitutes and extenders can play a critical role in underdeveloped societies where the population has limited access to quality protein. There is not a single word about vegetarians, how they can maintain a healthy lifestyle without eating animal protein. This aspect should also be highlighted.
Please be consistent with the style of percentages: L121:9.46%, L126: 3-9%, L129: 8.0%, L130: 37.9%, L301: 40 %,
L184: what does it mean “appearance” here? Color intensity, homogeneity of color, color hue, etc. Or was that a liking variable? please clarify
L187: “was scored lower” does it mean liking, or texture liking or overall liking?
L179-196. This section should be reformulated completely. The authors mix the results coming from consumer sensory analyses and those coming from a trained panel.
L205: one of the reasons
In section 2.1 the authors collected the results of the last 15 years, but in section 2.2. the last 13 years are listed. Why did you choose a different time frame?
L219: define WHC
L219: what does it mean increased sensory properties?
L222: define TBARS
Section 2.2. authors should rearrange the whole section. They mix the discussion of physicochemical and sensor changes, making the section harder to understand. There is no clear golden line here.
L282: Tofu is capitalized
L348: “F.” should be italic
L346-439: it should be noted that nutritional value of edible insects heavily depends on the used species and the developmental stage. see: https://doi.org/10.1016/j.ifset.2019.01.016
L465-468: consumer beliefs should also be considered here: https://doi.org/10.3390/su12135352
These two aspects should also be inserted here:
Neophobia and its effects on food choice should also be mentioned here: 10.1016/j.foodqual.2014.07.008
What drives consumers when it comes to meat alternatives: 10.1016/j.foodqual.2016.03.011, 10.1016/j.foodqual.2017.02.017
Tables. Tables should be extended. One column should be dedicated to the physicochemical changes, an other one should list the sensory results of trained panels (e.g. the changes in the intensities of given attributes) and a third should list the results of consumer studies (e.g. the liking of the products).
General comments. Detailed description of the collection method of the papers included in this review. The topic is large with much more paper available out there. Therefore, it is needed to state which method the authors used when choosing the papers.
Round 2
Reviewer 4 Report
The authors addressed all the issues I raised. Not all the suggestions were accteped but the given explanations were clear and informative. The paper is on a high level of quality and well-written. I give a minor revision just only because of my last three comments. I truly think that this paper has the potential to become highly cited.
L196: As a sensory scientist, I need to mention that is should be noted here that trained panelists should not conduct hedonic tests. I understad that the authors are awavre of this, however, if it is not stated here, the reader would never now. Please help spreading the good practices.
L204: "...found that pork patties pork patties where..."
L213: sentence ending is missing.
The references section does not meet the journal's requirements. Years should be bold (e.g. refs 66, 67, etc), titles should not be capitalized (e.g. refs 74, 85).
Author Response
The authors addressed all the issues I raised. Not all the suggestions were accteped but the given explanations were clear and informative. The paper is on a high level of quality and well-written. I give a minor revision just only because of my last three comments. I truly think that this paper has the potential to become highly cited.
We really want to thank Reviewer 4 for the constructive revision to help us improve the manuscript. The changes have been applied as requested by the reviewer and can be observed in the track changes of the word document.
These are:
L196: As a sensory scientist, I need to mention that is should be noted here that trained panelists should not conduct hedonic tests. I understad that the authors are awavre of this, however, if it is not stated here, the reader would never now. Please help spreading the good practices.
The sentence now reads (lines 196-199): These sensory analyses were done by trained panellists on a 9-point hedonic scale and even though this practise should be avoided—hedonic analyses should always be carried out by non-trained panellists—it can give an idea of the sensorial properties of the product.
L204: "...found that pork patties pork patties where..."
This has been corrected and the repetition of pork patties is no longer in the text. Line 206.
L213: sentence ending is missing.
We have added an ending to the sentence and it reads like this (lines 213-216):Unfortunately, a limitation from pulses and legumes as extenders can be found on the allergenic potential of some proteins contained in soybean and peanuts that would restrict population access to these products (people with allergies) and would need proper labelling
And we have also added a similar sentence to cereals, as some of them are also allergens (lines 284-287): It is also important to highlight that cereals containing gluten (wheat, rye, barley, and oats) have allergenic potential that must be declared in the labelling.
The references section does not meet the journal's requirements. Years should be bold (e.g. refs 66, 67, etc), titles should not be capitalized (e.g. refs 74, 85).
Reviewer is right and although we were using EndNote with the MDPI style some of the references are not correctly shown. These have been corrected as follows:
Lines year corrected to bold: 568, 574, 659,661,694,702,761,764,865
Lines corrected capital letters to small: 775-777,781-783,817-818